# Changes Caused by Low Doses of Bisphenol A (BPA) in the Neuro-Chemistry of Nerves Located in the Porcine Heart

**DOI:** 10.3390/ani11030780

**Published:** 2021-03-11

**Authors:** Krystyna Makowska, Slawomir Gonkowski

**Affiliations:** 1Department of Clinical Diagnostics, Faculty of Veterinary Medicine, University of Warmia and Mazury in Olsztyn, Oczapowskiego 14, 10-957 Olsztyn, Poland; 2Department of Clinical Physiology, Faculty of Veterinary Medicine, University of Warmia and Mazury in Olsztyn, Oczapowskiego 13, 10-957 Olsztyn, Poland; slawomir.gonkowski@uwm.edu.pl

**Keywords:** neuropeptide Y, heart innervation, bisphenol A, domestic pig

## Abstract

**Simple Summary:**

Bisphenol A (BPA) is a substance commonly used in the plastics industry, which is a part of many everyday items. It may leach from plastics and penetrate food, water, soil and air. It is known that BPA negatively affects living organisms. It impairs the functions of the intestine, neurons, reproductive organs, endocrine glands and immune cells. Previous studies have also reported that BPA negatively influences the cardiovascular system, leading to heart arrhythmia, intensification of atherosclerosis, blood hypertension and increased risk of a heart attack. However, many aspects of the influence of BPA on the heart are still poorly understood. One of these aspects is the BPA impact on heart innervation. Therefore, this article aimed to investigate the influence of low doses of BPA on the number of nerves containing selected active substances taking part in neuronal stimuli conduction located in the porcine heart apex. The results indicate that even relatively low doses of BPA are not neutral to the cardiovascular system, because they affect the neurochemical characterization of nerves in the heart. These changes may underlie the negative effects of BPA on the heart.

**Abstract:**

Bisphenol A (BPA) contained in plastics used in the production of various everyday objects may leach from these items and contaminate food, water and air. As an endocrine disruptor, BPA negatively affects many internal organs and systems. Exposure to BPA also contributes to heart and cardiovascular system dysfunction, but many aspects connected with this activity remain unknown. Therefore, this study aimed to investigate the impact of BPA in a dose of 0.05 mg/kg body weight/day (in many countries such a dose is regarded as a tolerable daily intake–TDI dose of BPA–completely safe for living organisms) on the neurochemical characterization of nerves located in the heart wall using the immunofluorescence technique. The obtained results indicate that BPA (even in such a relatively low dose) increases the number of nerves immunoreactive to neuropeptide Y, substance P and tyrosine hydroxylase (used here as a marker of sympathetic innervation). However, BPA did not change the number of nerves immunoreactive to vesicular acetylcholine transporter (used here as a marker of cholinergic structures). These observations suggest that changes in the heart innervation may be at the root of BPA-induced circulatory disturbances, as well as arrhythmogenic and/or proinflammatory effects of this endocrine disruptor. Moreover, changes in the neurochemical characterization of nerves in the heart wall may be the first sign of exposure to BPA.

## 1. Introduction

Bisphenol A (BPA) (Figure 1) is a synthetic chemical widely used in the plastics industry since the 1950s, because items made using this substance are relatively cheap, strong and easy to use [1,2]. BPA is a component of many things for everyday use, including bottles, toys, containers, furniture and many others. BPA is also present in thermal paper and even dental fillings [2]. It is known that BPA may leach from plastics and penetrate food and water. Due to the widespread use of BPA, this substance pollutes surface water, air and soil all over the world [3]. BPA also penetrates living organisms through the gastrointestinal tract, skin and respiratory system, as evidenced by the presence of BPA in the blood, urine and tissues of humans and other organisms [3,4]. In living organisms, BPA binds to the estrogen receptors and, as a strong endocrine disruptor, negatively affects various internal organs and systems [1,3].

Previous studies have reported that BPA, among others, affects the central and peripheral nervous system, impairing ion transport, synaptic activity and changing the neurochemical characterization of neuronal cells [5]. Moreover, BPA disturbs the endocrine system, influences the functions of the reproductive organs, causes pathological changes in the gastrointestinal tract and modulates immune cell activity [3,6,7,8]. Previous investigations have also reported correlations between exposure to BPA and the risk of neurodegenerative diseases, obesity, diabetes and neoplasms [3,9,10,11,12].

One of the internal systems subject to the adverse impact of BPA is the cardiovascular system. Previous experimental studies on various animal species have reported that BPA affects the cardiac electrical rhythm and causes “triggered activities” in cardiomyocytes, which leads to the development of cardiac arrhythmia [13,14]. Other studies have shown that BPA intensifies atherosclerosis processes in the cardiovascular system [15,16] and oxidative stress reactions in cardiomyocytes [17]. In turn, the authors’ previous study showed that BPA in high doses affects the neurochemical characterization of nerves located in the heart wall [18]. Moreover, it is known that BPA increases the risk of perimyocarditis and fibrosis after viral infections [19]. The influence of BPA on the cardiovascular system has been noted not only under the direct impact of BPA, but also in the offspring of mothers exposed to this substance during pregnancy. Changes noted in the offspring have included disturbances in the development of the cardiovascular system, cardiac hypertrophy and a higher risk of morphological changes in the heart and atherosclerosis in later life [20,21,22].

Observations concerning the influence of BPA on the cardiovascular system conducted during experimental investigations have been confirmed by epidemiological studies on humans. These studies have confirmed the correlations between the degree of exposure to BPA and the risk of various cardiovascular diseases, including coronary heart disease, myocardial infarction and heart angina [23]. It is also known that higher exposure to BPA may result in more frequent cases of blood hypertension and heart attacks [24,25] as well as peripheral arterial diseases and a reduction of the thickness of the arterial wall [26,27].

Nevertheless, many aspects connected with the impact of BPA on the cardiovascular system are not clear. One of them is the influence of BPA on the neurochemical characterization of nerves supplying the heart. Previous studies have indicated that changes in the expression of active substances in the nerves within the heart wall may appear under the impact of higher doses of BPA [18]. On the other hand, whether changes in the neurochemical characterization of nerves in the heart wall are also caused by lower doses of BPA remains unclear.

Therefore, the present study investigated the influence of BPA in a dose of 0.05 mg/kg body weight/day (b.w./day) on heart innervation. Although according to previous studies this dose may be considered as a low dose of BPA [28], in many countries of the world it has been established as a tolerably daily intake or reference dose of BPA for humans which is completely safe [29]. However, the European Food Safety Authority (EFSA) temporarily reduced the TDI for BPA to 4 μg/kg b.w./day because a dose of 0.05 mg/kg b.w./day may cause changes in the immune system, although the final decision depends on further studies [30]. It is also known that the dose used in the present study is higher than the average exposure of humans during everyday life [31] but humans may be exposed to even higher concentrations of BPA in some situations. For example, previous studies have reported that the amount of BPA leached from dental fillings may amount to 30 mg/day [2,32].

It should be pointed out that the domestic pig is increasingly considered to be a good animal model for studies concerning pathological processes within the nervous system supplying the internal organs in humans. This is because the organization of the innervation of internal organs and neurochemical organization of the autonomic neurons in this species are similar to those noted in humans [33,34]. Therefore, the results of the present study may be the first step to a better understanding of mechanisms connected with the BPA impact on heart innervation in humans.

## 2. Materials and Methods

The present investigation involved ten female juvenile pigs of the Piétrain × Duroc breed at the age of eight weeks. All procedures within the framework of the experiment were approved by the Local Ethics Committee responsible for experiments with the use of animals in Olsztyn (Poland) (decision numbers 28/2013 of 22 May 2013 and 65/2013/DLZ of 27 November 2013). During the experiment, animals were kept in standard conditions appropriate for species and age. They were fed twice a day with commercial standard feed for piglets and had unlimited access to drinking water. The animals’ care during the experiment was previously described by Makowska et al. [18].

The animals were randomly divided into two groups of five animals in each: control (C) and experimental (Ex) group. All animals received capsules for 28 days during the morning feeding. Control animals received empty capsules and pigs in the Ex group received capsules filled with BPA at a dose of 0.05 mg/kg body weight (b.w.)/day. All animals were weighed once a week to establish the exact dose of BPA. 

After 28 days of the administration of capsules, all animals were euthanized. For this purpose, the animals were treated with Stresnil (Janssen, Beerse, Belgium, 75 μL/kg of b.w., given intramuscularly) and after about 30 min with an overdose of sodium thiopental (Thiopental, Sandoz, Kundl, Austria, given intravenously).

Immediately after death, the heart apex was collected from each animal. Fragments of the hearts were fixated in 4% buffered paraformaldehyde (pH 7.4) for 1 h at room temperature (rt) and rinsed in phosphate buffer for three days (storage at 5 °C, the buffer was changed every day). After this period, the tissues were put into 18% phosphate-buffered sucrose. They were stored at 5 °C for further studies (not less than three weeks). Fragments of hearts were then frozen at −22 °C, cut into 14-μm-thick sections using a cryostat (HM 525, Microm International, Dreieich, Germany) and mounted on microscopic slides, which were stored at −22 °C.

Tissues were subjected to a single immunofluorescence technique used previously by Szymanska et al. [35]. In this method, commercial antibodies against neuronal factors were studied as well as secondary antibodies conjugated with appropriate fluorochromes (Table 1). 

The individual stages of single immunofluorescence labeling (all performed at room temperature) are presented in Figure 2. To eliminate non-specific labeling typical tests of the specificity of primary antibodies were made up in this experiment. These were preabsorption, omission and replacement tests, and the application of these tests completely eliminated non-specific labeling in the heart apex.

The evaluation of labeled heart fragments was made up under an Olympus BX51 microscope (Olympus, Tokyo, Japan) equipped with epi-fluorescence and appropriate filter sets. To determine the density of nerve fibers containing particular neurochemical factors, the counting of such fibers per microscopic observation field (0.1 mm^2^) was performed. Fibers were counted in four slices of the heart appendix from each experimental animal and five microscopic fields were randomly included in the study from each slice. Thus, nerve fibers immunoreactive to each neuronal factor studied were counted within 20 microscopic observation fields derived from each animal. To prevent double-counting the same nerves, the slices of the heart apex were placed at least 100 µm one from another. Data were pooled and presented as mean ± SEM. The statistical analysis was conducted with a *t*-Student test using Statistica 13.3 software (StatSoft Inc., Tulsa, OK, USA). The results were considered statistically significant at *p* < 0.05.

## 3. Results

During this investigation, under physiological conditions, nerves immunoreactive to neuropeptide Y (NPY), vesicular acetylcholine transporter (VAChT—used here as a marker of cholinergic—parasympathetic nerves), tyrosine hydroxylase (TH—a marker of adrenergic sympathetic nerves) and/or substance P (SP) were found in the porcine heart apex (Table 2, Figure 3, Figure 4, Figure 5, Figure 6, Figure 7 and Figure 8). 

Neuropeptide Y (NPY), vesicular acetylcholine transporter (VAChT), tyrosine hydroxylase (TH), substance P (SP), cocaine- and amphetamine-regulated transcript peptide (CART) or calcitonin gene-related peptide (CGRP). Statistically significant (*p* ≤ 0.05) and highly statistically significant (*p* ≤ 0.001) differences between control animals (C) and animals receiving BPA (Ex) are marked with *. 

In control animals, the most numerous were nerves containing TH (Figure 7(IIa)). Their average number was 27.22 ± 0.14 fibers per observation field. Nerves immunoreactive to VAChT (Figure 7(Ia)) and/or NPY (Figure 7(IIIa)) were slightly less numerous. Their average number amounted to 21.02 ± 0.36 and 18.76 ± 0.19 fibers per observation field, respectively (Table 2). In turn, the density of nerves immunoreactive to SP was very low (Figure 8(Ia)). The average number of such nerves was only 0.87 ± 0.05 fibers per observation field. During the present study, nerve fibers immunoreactive to cocaine and amphetamine-regulated transcript peptide (CART) (Figure 8(IIIa)) or calcitonin gene-related peptide (CGRP) (Figure 8(IIa)) were not observed in the heart apex.

As regards the morphological properties of nerves immunoreactive to particular substances, fibers containing VAChT (Figure 7(Ia)) and/or TH (Figure 7(IIa)) were very thin, short and delicate and showed a rather low level of immunoreactivity. Nerves immunoreactive to NPY (Figure 7(IIIa)) and/or SP (Figure 8(Ia)) were also short, but they were a little thicker and better visible.

During the present study, it was found that the dosage of BPA at the level of 0.05 mg/kg body weight (b.w.)/day may cause changes in neurochemical characterization of nerves located in the heart apex (Table 2, Figure 7 and Figure 8), although experimental animals did not show any clinical symptoms of exposure to BPA. The most visible changes were observed for nerves immunoreactive to TH, whose average number increased after the administration of BPA to 34.45 ± 0.57 fibers per observation field (Table 2, Figure 7(IIa,IIb)). A BPA-induced increase in population size was also noted for nerves containing NPY (Figure 7(IIIa,IIIb)) and/or SP (Figure 8(Ia,Ib)), but fluctuations were less visible than those noted for TH-positive fibers. In particular, the average number of nerves containing NPY and/or SP after treatment with BPA amounted to 22.19 ± 0.49 and 1.23 ± 0.09 fibers per observation field, respectively (Table 2). Contrary to nerves containing TH, NPY and/or SP, BPA-induced changes in the number of VAChT positive nerves were not observed during the present study (Table 2, Figure 7(Ia,Ib)). The average number of VAChT—immunoreactive nerves in animals under the impact of BPA amounted to 22.26 ± 0.52 fibers per observation field and this value was not statistically significantly different from the number of VAChT—positive nerves observed in the control animals (Table 2). Moreover, in pigs receiving BPA, similarly to the control pigs, nerves immunoreactive to CART (Figure 8(IIIb)) and/or CGRP (Figure 8(IIb)) were not observed in the porcine heart apex during the present investigations. In the current experiment, it was found that administration of BPA influences the morphological properties of nerves containing TH (Figure 7(IIb)) and/or NPY (Figure 7(IIIb)). These nerves after treatment with BPA were slightly thicker and more visible in comparison to fibers observed in the control animals. Moreover, fibers containing NPY in animals receiving BPA often formed bundles composed of several fibers (Figure 7(IIIb)). On the other hand, administration of BPS did not change the morphology of nerves immunoreactive to VAChT (Figure 7(Ib)) and/or SP (Figure 8(Ib)).

## 4. Discussion

The most numerous populations of nerves noted in the present study contained VAChT or TH. There is nothing unusual about this, taking into account that VAChT is considered to be a marker of neurons synthesizing acetylcholine and TH is a marker of neurons synthetizing noradrenaline, which are the main neurotransmitters in the parasympathetic and sympathetic nervous system, respectively. The key roles of these neurotransmitters in regulation of the heart functionality are relatively well known. 

For many decades, it is commonly known that activation of cholinergic innervation causes, among others, a reduction of heart rate, contractile strength and velocity of stimuli conduction within the electrical conduction system of the heart, while the activation of adrenergic innervation has the opposite effects [36,37,38,39,40].

In the current experiment, the third-most abundant was the NPY-containing fiber population. Previous studies have described the presence of this neuropeptide in the heart in both postganglionic sympathetic nerves originating from stellate ganglion [41,42] and in parasympathetic fibers in the processes of neuronal cells located in the intracardiac ganglia [43]. NPY may act on the cardiomyocytes, causing their stimulation or inhibition and such opposing effects result from the fact that NPY may act on the cardiac muscle through various types of receptors [44,45,46]. Moreover, NPY is known as a substance that reduces acetylcholine release [47] and therefore attenuates vagal bradycardia [48], as well as shows pro-arrhythmic activity by direct impact on the ventricular electrophysiology [49]. Apart from the influence on cardiomyocyte contraction, NPY is known as an important trophic factor taking part in heart development [50] and a substance that influences blood vessels and improves myocardial perfusion [51]. This neuropeptide also participates in the development of some heart and cardiovascular diseases, including hypertension, myocardial infarction and chronic heart failure [52].

The next population of fibers noted in the present experiment were nerves containing SP. Their numbers observed in this study were relatively low, although it is known that SP in the heart may play various functions. First of all, SP, which is considered to be a key, classic sensory neurotransmitter in mammals [53], participates in the conduction of sensory and pain stimuli from the heart to the central nervous system [54]. Previous studies have also shown that SP may participate in many other regulatory processes in the heart. In particular, SP influences cardiomyocyte activity through the activation of cholinergic innervation and causes bradycardia [55,56]. Moreover, SP regulates the blood flow in vessels located in the heart wall (the character of this impact depends on gender) [57,58] and may stimulate the growth of cardiomyocytes [59] and influence the heart mast cells, stimulating the release of pro-inflammatory factors [60,61,62]. It is also known that SP participates in mechanisms connected with various heart diseases, including viral and parasitic infections, heart ischemia and cardiac fibrosis [54,59].

In this experiment, there was no presence of nerve fibers containing CART and/or CGRP, although previous studies have described both of these substances in the intracardiac nervous structures [63,64,65,66]. The absence of CGRP positive nerves is especially surprising because this substance in the light of previous studies is known as an important factor participating, among others, in the regulation of the blood flow in vessels located in the heart wall [67], showing protective effects on cardiomyocytes [67,68] and increasing the heart rate [69]. On the other hand, the present results concerning the absence of nerves immunoreactive to CART and/or CGRP in the porcine heart apex are in agreement with the previous observation [18].

In this experiment, changes in the neurochemical characterization of nerves located in the heart apex under the impact of BPA were also observed. These observations indicate that not only high doses of BPA affects the nerves located in the cardiac wall, as has been previously described [18], but relatively low doses of this endocrine disruptor may also change the expression of active substances in nervous structures supplying the heart. Therefore, the obtained results suggest that even low doses of BPA, which in some countries of the world are described as safe for humans and animals [28,29], are not completely harmless. This observation is in agreement with previous studies, in which changes in the autonomic innervation of various internal organs have been observed under the impact of this dosage of BPA [35,70,71].

The precise processes responsible for changes in the immunoreactivity of nerves, observed in this experiment are not clear, but adrenergic innervation of the heart seems to play a more important role in processes induced by BPA than the cholinergic structures. This is evidenced by a marked increase in the number of fibers containing TH and no changes in the population size of nerves immunoreactive to VAChT were noted in the present study.

It is most likely that the observed changes are associated with the impact of BPA on cardiomyocyte activity and heart rate. Previous studies have reported that BPA is a factor, which induces heart arrhythmia and arrhythmogenic effects of this endocrine disruptor are especially visible in females [13,14,72]. Moreover, it is known that even low doses of BPA induce heart cycle disorders [14] and cellular mechanisms of BPA-induced arrhythmia are connected with alterations in transmembrane calcium ion transport in ventricular cardiomyocytes [73], resulting from rapid estrogen receptor-mediated reactions [74]. Moreover, it is known that arrhythmic processes caused by BPA are regulated by the balance in the expression of various types of estrogen receptors, by which this endocrine disruptor (that resembles estrogen in its structure) acts on the heart [74].

The thesis that BPA-induced arrhythmia is connected with changes in neurochemical characterization of nerve fibers noted in this experiment is likely. In particular, one group of fibers whose number increased following the administration of BPA were nerves immunoreactive to NPY, described previously as a pro-arrhythmic factor [49]. The visible increase in the number of nerves containing NPY and TH may also result from the fact that exposure to BPA results in a slowed spontaneous heart rate [75]. It is known that both adrenergic innervation, as well as nerves containing NPY, may increase the heart rate [38,46] and the increase in the number of nerves immunoreactive to these substances aim at homeostasis maintenance in cardiomyocyte activity.

It cannot be excluded that changes noted in the present study may result from other activities of BPA. One of them may be connected with the relatively well-known direct neurotoxic impact of this endocrine disruptor on the nervous structures [76] and changes in the number of nerves containing a particular active substance may be a manifestation of adaptive and/or neuroprotective reactions aimed at the proper functioning of the nervous system under the toxic influence of BPA. A similar situation was observed in other parts of the nervous system [35,70,71,77]. Among the substances studied in this investigation, NPY is a factor with the best-known protective properties. NPY shows neuroprotective effects by trophic support of neuronal cells, inhibition of excitotoxicity and stabilization of intraneuronal calcium homeostasis [78]. The latter property of NPY may be the key reason for the noted changes because the toxic activity of BPA is based to a large extent on disturbances of calcium ion transport through the cell membrane [79,80,81]. Similarly to NPY, noradrenaline (during this study a visible increase in the number of adrenergic nerves was noted) and substance P (the observed increase in the number of SP-positive nerves was rather low) also showed neuroprotective activity [82,83].

Fluctuations noted in this experiment may also result from the strong pro-inflammatory properties of BPA. It is known that this endocrine disruptor modifies the immunological system, which results in the stimulation of pro-inflammatory cytokine production and a decrease in the population size of T and B cells [84,85]. Such activity of BPA may cause an increase in the number of nerves immunoreactive to NPY and/or SP. Both of these substances are known as pro-inflammatory factors. NPY modulates inflammatory processes through various reactions, including inhibition of lymphocyte proliferation, granulocyte oxidative burst, influence on the activity of natural killer cell and the production of pro-inflammatory cytokines [86]. Similar properties are shown by SP, which (among others) intensively stimulates the synthesis of tumor necrosis factor alpha and interleukins, including IL-l, IL-6 and IL-8 [60,87,88,89]. Admittedly, the dosage of BPA used in the present experiment was low and animals did not show any symptoms of inflammatory processes, but it cannot be excluded that fluctuations in the neurochemical characterization of nerve fibers and an increase in the number of nerves containing factors showing pro-inflammatory activity are the first sign of BPA-induced subclinical inflammation. 

## 5. Conclusions

The results obtained in this experiment indicate that even relatively low dosages of BPA (considered safe for living organisms) given for a short period (28 days) affect neurochemical properties of nerves placed in the heart apex. Therefore, it stands to reason that this dose is not neutral for humans and animals, and changes in neurochemical properties of nerve fibers are the first signs of intoxication with BPA. Mechanisms of BPA-induced fluctuations in the neurochemical profile of nerves in the heart wall are not clear. Most likely, the changes noted in the present study result from the pro-arrhythmic activity of this endocrine disruptor, but they may also be connected with neurotoxic and/or proinflammatory properties of BPA. However, further investigation is needed to clarify all aspects of the impact of BPA on heart innervation.

## Figures and Tables

**Figure 1 animals-11-00780-f001:**
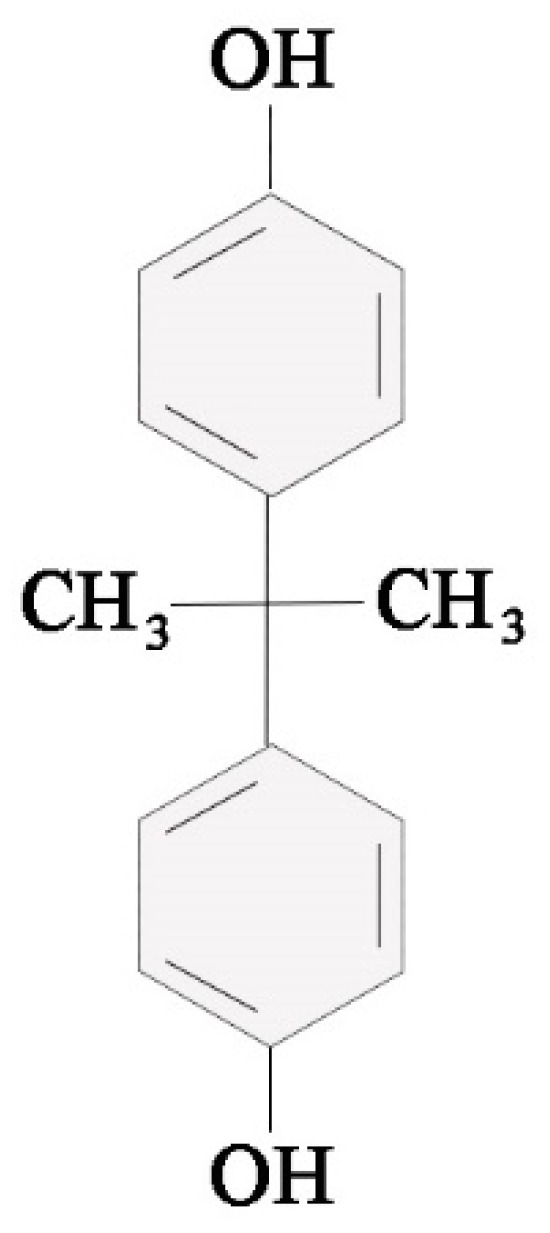
Scheme of the chemical formula of bisphenol A (BPA).

**Figure 2 animals-11-00780-f002:**
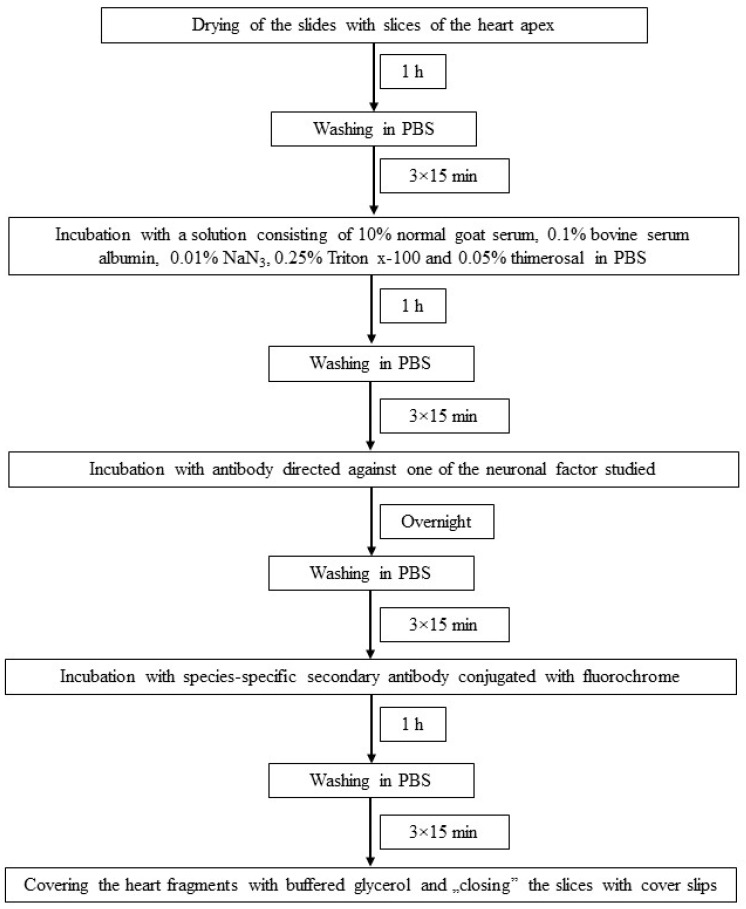
The scheme of immunofluorescence technique used in the present investigation.

**Figure 3 animals-11-00780-f003:**
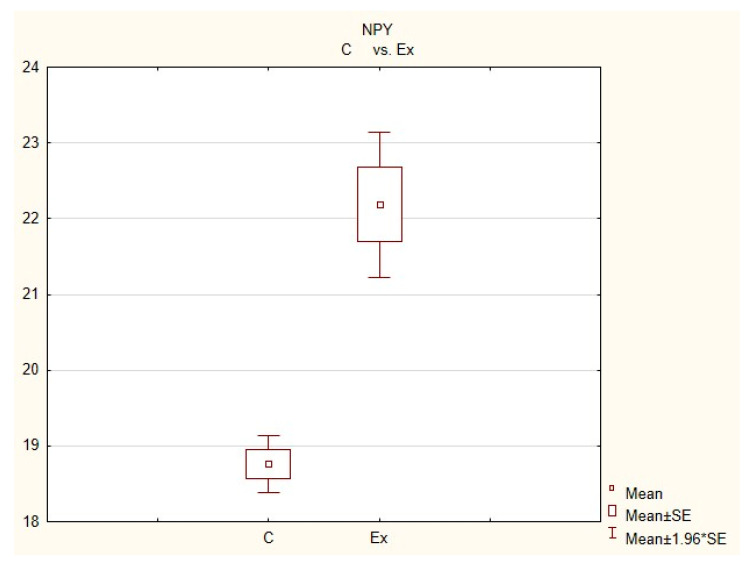
The average number of nerves immunoreactive to neuropeptide Y(NPY) in the porcine heart apex in physiological conditions (C) and after administration of bisphenol (BPA) at a dose of 0.05 mg/kg body weight/day (Ex).

**Figure 4 animals-11-00780-f004:**
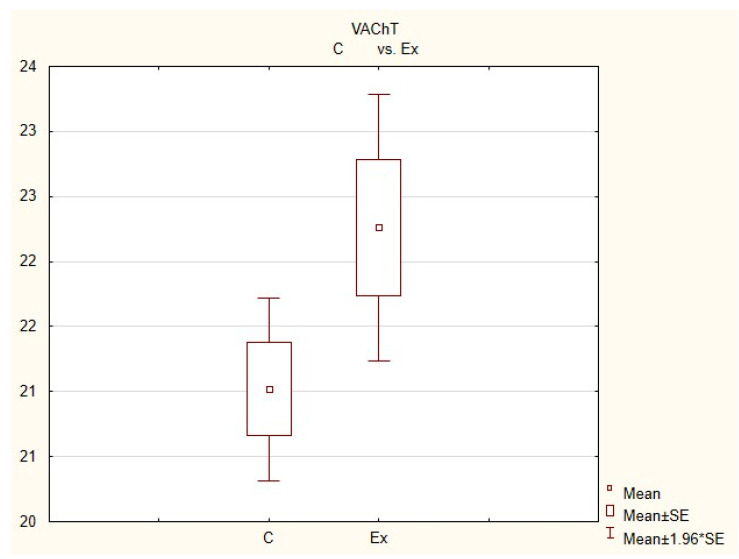
The average number of nerves immunoreactive to vesiculr acetylcholine transporter (VAChT) in the porcine heart apex in physiological conditions (C) and after administration of bisphenol (BPA) at a dose of 0.05 mg/kg body weight/day (Ex).

**Figure 5 animals-11-00780-f005:**
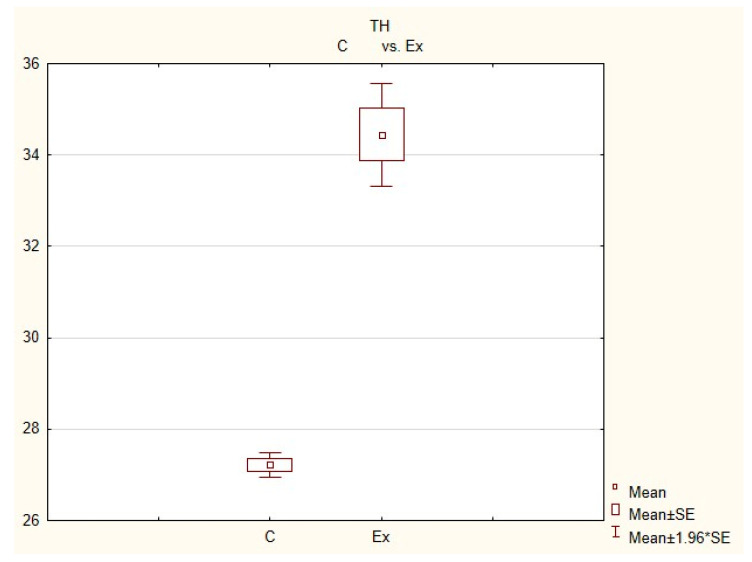
The average number of nerves immunoreactive to tyrosine hydroxylase (TH) in the porcine heart apex in physiological conditions (C) and after administration of bisphenol (BPA) at a dose of 0.05 mg/kg body weight/day (Ex).

**Figure 6 animals-11-00780-f006:**
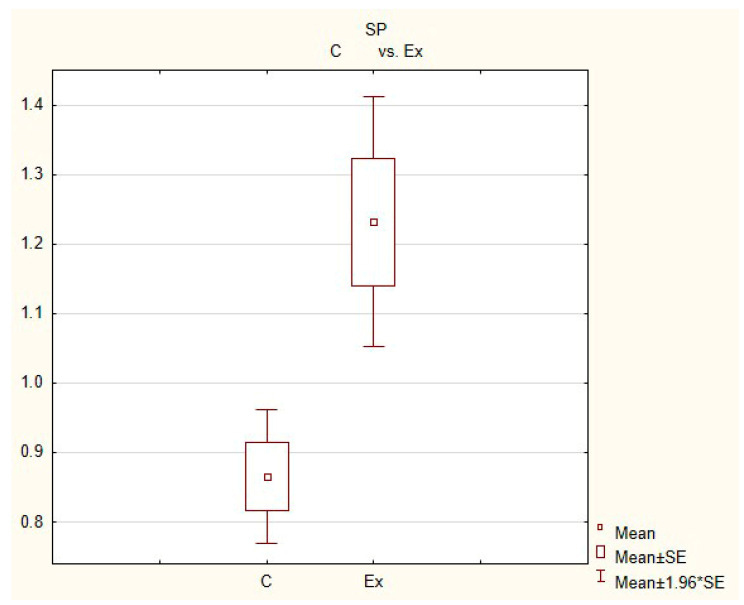
The average number of nerves immunoreactive to substance P (SP) in the porcine heart apex in physiological conditions (C) and after administration of bisphenol (BPA) at a dose of 0.05 mg/kg body weight/day (Ex).

**Figure 7 animals-11-00780-f007:**
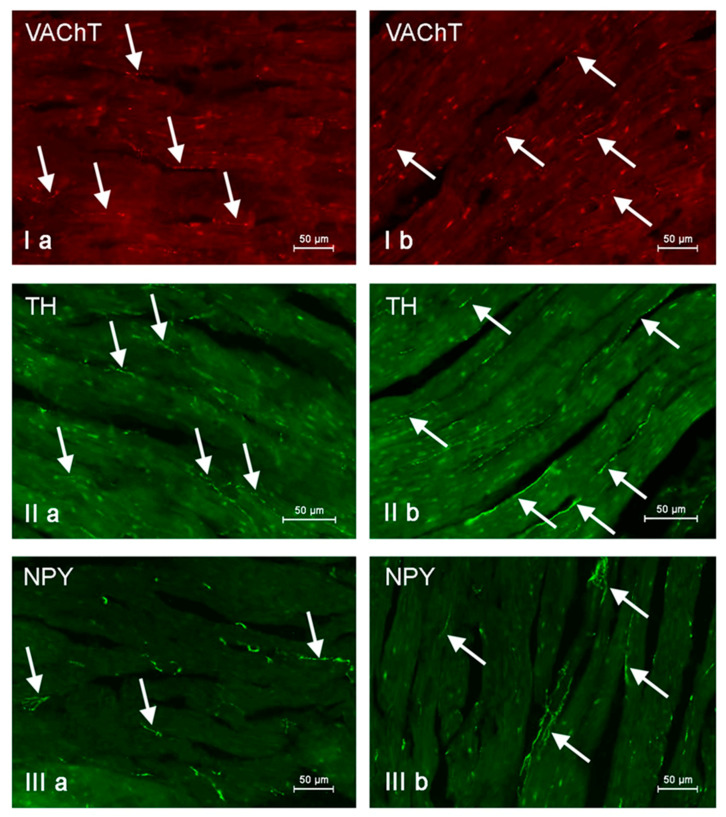
Nerve fibers immunoreactive to vesicular acetylcholine transporter (VAChT), tyrosine hydroxylase (TH) and neuropeptide Y (NPY) in the porcine heart apex under physiological conditions (**a**) and after administration of bisphenol A in a dosage of 0.05 mg/kg body weight/day (**b**). Nerves containing particular active substances are indicated with arrows.

**Figure 8 animals-11-00780-f008:**
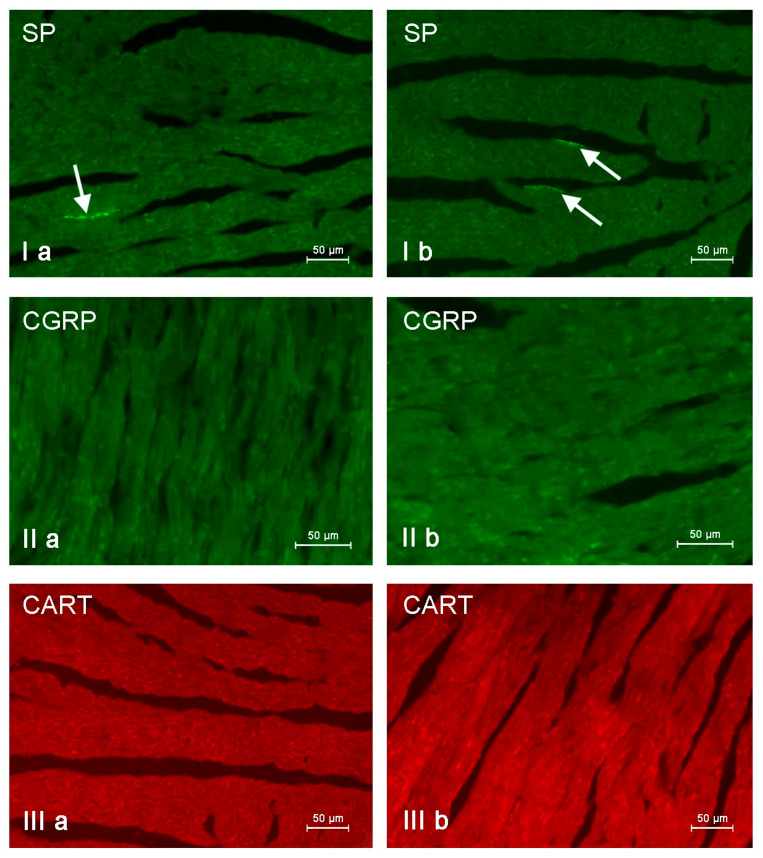
Nerve fibers immunoreactive to substance P (SP), calcitonin gene-related peptide (CGRP) and cocaine- and amphetamine-regulated transcript peptide (CART) in the porcine heart apex under physiological conditions (**a**) and after administration of bisphenol A in a dosage 0.05 mg/kg body weight/day (**b**). Nerves containing particular active substances are indicated with arrows.

**Table 1 animals-11-00780-t001:** Antibodies used in the present experiment.

**Primary Antibodies**					
**Antigen**	**Catalogue No.**	**Host Species**	**Working Dilution**	**Supplier**	
CART	1-003-61	Rabbit	1:8000	Phoenix Pharmaceuticals, INC, Belmont, CA, USA	
CGRP	T-5027	Guinea pig	1:1600	Peninsula, San Carlos, CA, USA	
NPY	NA 1115	Rabbit	1:2000	Biomol, Hamburg, Germany	
SP	8450-0505	Rat	1:1000	Bio-Rad (AbD Serotec), Kidlington, UK	
TH	MAB 318	Mouse	1:400	Millipore, Warszawa, Polska	
VAChT	H-V006	Rabbit	1:2000	Phoenix Pharmaceuticals	
**Secondary Antibodies**					
**Reagent**				**Working Dilution**	**Supplier**
Alexa Fluor 488 conjugated goat anti-rat IgG				1:1000	Invitrogen, Carlsbad, CA, USA
Alexa Fluor 488 conjugated goat anti-mouse IgG				1:1000	Invitrogen
Alexa Fluor 488 conjugated goat anti-guine pig IgG				1:1000	Invitrogen
Alexa Fluor 546 conjugated goat anti-rabbit IgG				1:1000	Invitrogen

**Table 2 animals-11-00780-t002:** The average number of nerves immunoreactive to particular neuronal factors (per observation field—0.1 mm^2^) in the porcine heart apex in physiological conditions (C) and after administration of bisphenol (BPA) a at a dose of 0.05 mg/kg body weight/day (Ex). The statistically significant differences between control and experimental groups were marked with *.

	NPY		VAChT		TH		SP		CART		CGRP	
	C	Ex	C	Ex	C	Ex	C	Ex	C	Ex	C	Ex
Animal 1	18.73	23.95	21.80	20.85	26.73	36.10	0.80	1.38	0	0	0	0
Animal 2	18.20	21.65	21.70	22.73	27.53	34.80	0.95	1.45	0	0	0	0
Animal 3	18.53	22.20	19.83	22.13	27.28	33.35	0.73	0.93	0	0	0	0
Animal 4	19.30	21.00	20.73	21.65	27.43	35.00	1.00	1.15	0	0	0	0
Animal 5	19.05	22.13	21.03	23.95	27.13	32.98	0.85	1.25	0	0	0	0
Mean ± SEM	18.76 0.49 *	22.19 0.49 *	21.02 0.36	22.26 0.52	27.22 0.14 *	34.45 0.57 *	0.87 0.05 *	1.23 0.09 *	0	0	0	0

## Data Availability

Data is contained within the article.

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
