# Peer review of "Changes Caused by Low Doses of Bisphenol A (BPA) in the Neuro-Chemistry of Nerves Located in the Porcine Heart"

_animals, 2021, doi:10.3390/ani11030780_

Round 1

Reviewer 1 Report

Comments to the Author

The report by Makowska and Gonkowski is an interesting study focusing on the changes caused by low doses of Bisphenol A (BPA) in neurochemistry of nerves supplying the porcine heart. These observations suggest that changes in the heart innervation can be at the root of BPA-induced circulatory disturbances, as well as arrhythmogenic and/or proinflammatory effects of this endocrine disruptor. The results are novelty and should be shared with the community. Although I have some concerns on the manuscript:

  1. It would be good to add the chemical formula scheme of BPA to the manuscript
  2. In the tables the word „average” should be replaced with “mean”
  3. Did any symptoms appear in the animals during the experiment? Some changes in the weight gain or appetite/ water consumption ?
  4. In the materials and methods section the previous study of the authors regarding the administration of high doses of BPA should be if it was one experiment
  5. Please check the manuscript for double spaces and remove them where they appear
  6. After the short “min” shouldn’t be a dot
  7. It would be nice to show the results also in diagrams, not only in tables, then changes between groups would be better visible

Author Response

The authors thank for the revision, which allows to improve the manuscript.

The scheme of BPA has been added (Figure 1).

In the tables the word „average” was replaced with “mean”.

There was no symptoms in any group of animals, the weight gain, appetite and water consumptions were same (there were no statistically significant changes) in control and experimental groups. The assumption was to use small doses of BPA which shouldn’t cause any symptoms and in fact during the experiment none symptoms were noticed in the animals.

In the materials and methods a citation of the previous study was added.

The manuscript was checked for double spaces and they were removed when they appeared.

A dot after the short “min” was deleted.

The graphs of the results were added to the Results section.

The authors hope that improvements will allow to public the manuscript in journal “Animals”.

Reviewer 2 Report

The study by Makowska investigated the impact of BPA in a dose of 0.05 mg/kg body weight/day on the neurochemical characterization of nerves located in the heart wall using immunofluorescence technique. They found that the changes in the heart innervation can be at the root of BPA-induced circulatory disturbances, as well 35 as arrhythmogenic and/or proinflammatory effects of this endocrine disruptor. Overall, the study was well presented and organised. 

A few minor issues:

  1. English language and style shall be improved.
  2. Institutional Review Board Statement: statement shall be provided as this study used pigs.
  3. Data Availability Statement shall be provided.  

Author Response

The authors thank for the insightful review, which allows to improve the manuscript.

English language and style were corrected by a native speaker.

Institutional Review Board Statement: was provided.

Data Availability Statement was provided.

The authors hope that improvements will allow to public the manuscript in journal “Animals”